# Protocol for a qualiquantitative study of accessibility of sexual and reproductive health services among women with motor disabilities in Morocco

**Firdaous Zekaoui**[1]*, **Houssine Boufettal**[2], **Rachid Aboutaieb**[2]

**1** Sexual and Reproductive Health Laboratory LSSR, Faculty of Medicine and Pharmacy, Hassan II University, Casablanca, Morocco, **2** Faculty of Medicine and Pharmacy, Hassan II University, Casablanca, Morocco

* zekaoui.firdaous@gmail.com

## Abstract

### Introduction

Approximately 1.3 billion people worldwide live with a disability, underlining the need for equitable health care for this group is important. Inclusive health systems are consistent with global health priorities, respond to United Nations sustainable development goals and economically viable. Despite advancements in Morocco's health system and legal framework, significant disparities remains inaccessing sexual and reproductive health (SRH) services for women with disabilities. This study explores barriers and facilitators affecting access to SRH services for these women, aiming to provide recommendations for more inclusive services and improved health outcomes.

### Methods

Using a mixed approach, this study assessSRH access for women with motor disabilities in two cities in Morocco, selected for their socio-demographic, cultural and and healthcare diversity. Quantitative data will be generated through questionnaires administrered to 400 participants, while qualitative information will come from 20 to 30 interviews. Recruitment involves working with health professionals, NGOs and social media campaigns to ensure a diverse sample. Data analysis includes NVivo for qualitative data and descriptive/inferential statistics for quantitative data.

### Conclusion

The study aims to explore the perspectives of women with disabilities and health professionals regarding SRH services. It will provide recommendations for making SRH services more equitable and inclusive, ultimately improving health out comes and uphold health rights for women with disabilities.

**Data Availability Statement:** No data sets were generated or analysed during the current study. All relevant data from this study will be made available upon study completion.

**Funding:** The author(s) received no specific funding for this work.

**Competing interests:** The authors have declared that no competing interests exist.

## Introduction

According to the World Report on Disability [1], around 1.3 billion people, or 16% of the world's population, have a disability. Achieving health equity for this significant segment of the population would contribute to global health and wellness goals. Investing in their inclusion in the health sector is a cost-effective strategy, as it supports progress towardsuniversal health coverage and the achievement of the 2030 Sustainable Development Goals (SDGs), particularly SDG 3. It also fosters improvements in their health through inclusive, cross-sectoral interventions.

The same analysis suggests that for every dollar spent on cancer prevention and control, with accessibility ensured for people with disabilities, nearly nine dollars could potentially be gained. These figures challenge the widely-held belief that integrating people with disabilities into the healthcare system is costly and difficult. On the contrary, research demonstrates that doing so is both financially sound and morally just.

It is therefore crucial to implement policies and procedures that guarantee health care access for all. Given the holistic approach to health, which encompasses all aspects, including sexual and reproductive health, ensuring access to sexual and reproductive health services is essential for promoting overall well-being and improving the quality of life for people with disabilities.

### Global background

The global commitment to addressing disability, with a focus on inclusive and accessible services, has been gaining momentum. A key milestone in this effort was the 2006 adoption of the United Nations Convention on the Rights of Persons with Disabilities, which affirmed the equal right of individuals with disabilities to health services, including sexual and reproductive health (SRH). This encompasses access to informational, educational, preventive, therapeutic, and supportive services, with an emphasis on eliminating discrimination and tailoring services to meet individual needs through inclusive and accessible policies.

Various international conventions and declarations further highlight the importance of equitable SRH services for all, including persons with disabilities. For instance, Sustainable Development Goal (SDG) 3.7 calls for "universalaccess to sexual and reproductive health-care services, including family planning, information and education," integrating SRH into national programs and strategies. The 2019 Nairobi Statement on Population and Development reaffirmed this by emphasizing humanrights, equality, and inclusion in SRH services, particularly for people with disabilities.

Additionally, the United Nations Committee on the Rights of Persons with Disabilities has provided specific recommendations on SRH for people with disabilities, and the World Health Organization (WHO) has highlighted the need for inclusion and equity in SRH service delivery for women with disabilities. The Global Partnership for Disability and Development (GPDD) advocates for laws and initiatives thataddress the unique needs of persons with disabilities, ensuring their full inclusion in development efforts. These collective actions reflect a robust global commitment to ensuring non-discriminatory and equitable access to SRH services for all individuals, including those with disabilities.

### Moroccan setting

In Morocco, approximately 2 million individuals live with a disability, representing 6.8% of the population [2]. Of these, 45.5% are aged between 15 and 59, and 51.4% are women. Motor disabilities are the most prevalent, accounting for 50.2% of all cases. Morocco is committed to upholding human rights principles, ensuring equitable access to healthcare for all, including

sexual and reproductive health (SRH) services, regardless of disability, gender, or age. As a signatory to various international agreements, Morocco has enacted numerous laws and policies to support the inclusion and rights of people with disabilities (PWDs):2011 Moroccan Constitution protects the rights of PWDs and promotestheir full participation in society, framework law No. 97-13focusing on defending and advancing the rights of PWDs, integrated public policy for the promotion of the rights of persons with disabilities aims to improve social and economic inclusion through initiatives such as enhancing service accessibility, ratification of the CRPD (2009) ensuring PWDs' access to healthcare, including SRH services, commitment to the Sustainable Development Goals (SDGs) to aligns with global efforts to support marginalized populations, health and disability plan 2015–2021 aims to ensure PWDs have access to health services through a rights-based approach, health Plan 2025 (Axis 10) focusing on improving health promotion for populations with special needs, and the new Development Model seeks to reduce healthcare inequalities, promoting social inclusion.

These efforts have led to improvements in Morocco's overall health and SRH outcomes [3], including longer life expectancy, reduced maternal and child mortality, increased contraceptive use, lower fertility rates, advancements in early cancer detection, and laws addressing infertility, such as Law 47–14. However, disparities persist, particularly regarding access to information and SRH services for PWDs [4]. A 2018 report by the Guttmacher-Lancet Commission highlighted that PWDs often face discrimination, harmful stereotypes, and a lack of specialized services tailored to their SRH needs [5].

## Disability and access to sexual and reproductive health in the Morocco

An important problem for 60% of people with disabilities (DP) in Morocco is access to health services, mainly due to lack of infrastructure and adequate transportation, as highlighted by the 2014 National Disability Survey [6]. In addition, a separate study on the perception of people with disabilities in Morocco [7] found that financial constraints and mobility difficulties are the main barriers to access to care. Also, studies on the sexual and reproductive health needs and perceptions of people with disabilities noted a weakness in expressed needs due to lack of accessibility and adequate communication related to these services [8, 9].

Another study examined the influence of socioeconomic, educational, and cultural factors on women'saccess to reproductive healthcare. It identified poverty and economic vulnerability as key obstacles to obtaining quality care, which can contribute to higher maternal and infant mortality rates [10]. Education was also found to enhance access to reproductive healthcare, though sexual education remains inadequately addressed in Morocco's educational system, limiting young people's understanding of reproductive health, especially in rural areas. The study also suggested that the lack of health infrastructure, along with limited awareness campaigns and reproductive health information, negatively affects women'shealthacross the country.

A UNICEF report [11]on children's rights further revealed that PWDs in Morocco and similar nations face significant challenges in accessing healthcare. This finding aligns with a comparative study by Trani et al. [12], which noted gaps in Morocco's social security system, leaving many potential beneficiaries uncovered. The most vulnerable groups, particularly the poorest and those in rural areas, face even greater difficulties. Discrimination and negative attitudes within the healthcare system further hinder access to essential services.

Access to healthcare also involves the quality of care provided once services are obtained. Shipman discussed the disparities between Morocco and the United States in terms of the quality and availability of medical care, specialist services, and research in university hospitals. However, he also introduced the concept of "social capital," emphasizing the importance of

community networks in accessing healthcare in relational societies like Morocco, and how such support systems can play a crucial role in enabling PWDs to receive necessary care [13].

## Aim and objectives

This study aims to provide insights into the barriers and facilitators affecting access to sexual and reproductive health services among women with physical disabilities in Rabat and Khemisset. By understanding the perspectives of both women with disabilities and healthcare professionals, the study will contribute to developing recommendations for inclusive and equitable SRH services, ultimately improving health outcomes and promoting the rights of women with disabilities.

## Research questions

This study is guided by the following primary research question: What are the barriers and facilitators affecting access to sexual and reproductive health (SRH) services for women with physical disabilities in Rabat and Khemisset, Morocco?

This central question is further supported by secondary inquiries that explore the broader context:

1. How does disability affect women's access to SRH services in Morocco?

2. To what extent do current SRH services meet the specific needs of individuals with disabilities in Morocco?

3. What strategies could be employed to enhance accessibility to SRH services for women with disabilities in Morocco?

## Materials and methods

### Study design

The study will utilize a mixed-methods sequential explanatory design [14], incorporating an empirical approach to gather data from women with physical disabilities in two cities within the Rabat-Salé-Kenitra region. This approach will involve two distinct phases: first, quantitative data collection, followed by qualitative data collection. Such a design is well-suited for an in-depth exploration of the barriers and facilitators impacting access to sexual and reproductive health (SRH) services for women with motor disabilities. The sequential structure enables a deeper understanding, with qualitative methods providing further insights into the quantitative findings.

### Methods and techniques

In the quantitative phase of the study, a cross-sectional design will be employed, using structured surveys to assess awareness, knowledge, attitudes, perceptions, and practices related to sexual and reproductive health (SRH) services among women with disabilities and healthcare workers. For the qualitative phase, a descriptive phenomenological approach [15] will be used to explore the needs, experiences, and perceptions of women with disabilities and healthcare providers regarding SRH services and interventions. Semi-structured interview guides for the qualitative phase were developed based on a review of existing literature and validated by a multidisciplinary team, including experts in SRH, disability studies, and medical informatics. Participant recruitment and data collection will commence on August 1, 2024, with recruitment expected to conclude by March 20, 2025.

This study was approved by the Committee of Ethics for Biomedical Research, Mohammed V University–Rabat,Faculty of Medicine and Pharmacy of Rabat,Faculty of Dental Medicine of Rabat, Reference CERB 47–24. Informed verbal consent would be obtained from all participants, as approved by the Committee of Ethics for Biomedical Research, University Mohammed V–Rabat. The decision to obtain oral consent was made to accommodate participants who have issues or reservations for written consent. The research team will provide detailed information on the purpose of the study, procedures and rights of participants, ensuring that they were understood before verbal consent was given. The confidentiality of participants will be respected throughout the research, and they will be informed of their right to withdraw at any time without penalty. This studywill follow ethical guidelines for research involving human subjects.

## Protocol adjustments and methodological considerations for addressing recruitment and logistical challenges in research

Throughout the study, modifications to the research protocol may be necessary, especially to overcome challenges related to participant recruitment. Adjustments, such as extending the data collection period and revising the interview schedule, may be implemented to address logistical issues and participant availability. A comprehensive account of the process, data collection methods, and any challenges faced will be included in the final report to enable future researchers to replicate the study with accuracy.

## Variables to be collected

**Dependent variables.**

- Utilization and access to sexual and reproductive health (SRH) services, including the frequency of service use, awareness of available services, satisfaction levels, barriers to access, facilitators identified, specific needs and experiences of women with motor disabilities

**Independent variables.**

- Type and severity of motor disability, age, socio economic status, geographical location, educational attainment, marital status.

**Contextual variables.**

- Health policies, strategies, and programs, available resources, stigma and discrimination.

**Health professional-related variables.**

- Knowledge and specific training, perceptions, attitudes, and beliefs and how healthcare professionals view and treat women with motor disabilities in the context of SRH.

**Health service-related variables.**

- Availability of services tailored to specific needs, physical accessibility of health care facilities and equipment.

## Recruitment

To maximize inclusivity, we will employ multiple recruitment strategies [16]. Women with disabilities will be recruited through healthcare professionals, as well as through collaboration with NGOs and local organizations and associations that focus on disability issues. Healthcare professionals working in rehabilitation and adaptation centers, physical rehabilitation medicine services, and Regional Integrated Orthopedic and Rehabilitation Centers willbeasked to identify potential participants and seek their consent to be contacted by the research team. These professionals will also share information about the study, along with a phone number and email address for self-referral.

Local organizations and associations working in the field of disability will also promote the studyusing the same materials. Social media campaigns targeting women with disabilities in the regions of Rabat and Khemisset will further enhance recruitment efforts. Recruitment posters and announcements on relevant forums will be utilized to reach a broader audience. Once contact has been established, a member of the research team will reach out to potential participants to provide a brief overview of the study and answerany questions theymight have. If the individualis interested in participating, they will receive a copy of the participant information. This information sheet will outline the purpose and nature of the study, as well as the ethical safeguards regarding data protection and privacy.

## Sampling

A combination of non-probability convenience sampling for the qualitative phase and stratified random sampling for the quantitative phase will be employed. The sampling process will proceed as follows:

**Qualitative phase (stratified and purposive sampling).**   For Women with Disabilities: To capture diverse experiences among women with physical disabilities, both stratified and purposive sampling techniques will be utilized.

Stratified Sampling (by place of residence): Participants will be selected based on their location, ensuring representation from both urban and rural areas.

Purposive Sampling: Women with mobility impairments and relevant experiences regarding access to sexual and reproductive health (SRH) services will be specifically targeted.

Snowball Sampling:After recruitment through stratified or purposive methods, participants will be encouraged to refer others who might also contribute valuable insights to the study.

For Healthcare Professionals in Public Primary Healthcare Facilities (PHFs) or Reference Centers for Reproductive Health (RCRHs): A random selection of healthcare facilities will be made from a list of primary healthcare centers. This selection will include four facilities in Rabat and four in Khemisset(two rural and twourban) providing SRH services, as well as two RCRHs in Rabat and one in Khemisset.

Healthcare professionals from these selected facilities will be directly contacted and invited to participate in semi-structured interviews. The interviews aim to explore their perspectives, needs, and experiences in providing SRH services to persons with disabilities.

For Institutional Leaders, Policymakers, and Stakeholders:Individualsfrom national and international NGOs, associations, policymakers, and managers involved in disability-related or SRH policies and services will be included.

Purposive Sampling: National policymakers, representatives from international organizations, and national NGOs engaged in SRH services or disability-related policies will be selected. One representative from each institution will be contacted via email, phone, or mail and invited to participate in individual interviews, with the schedule determined based on their availability.

**Quantitative phase (stratified random sampling).**   To enhance the representativeness of the target population and enable more reliable generalizations, a stratified random sampling method will be employed during the quantitative phase of the study. This method involves dividing the target population into distinct subgroups, or strata, based on relevant characteristics, and then randomly selecting samples from each stratum.

*Stratification Criteria.* The following criteria will guide the formation of strata:

For Womenwith Disabilities:

Residence (Urban vs. Rural): This stratification will help capture differences in access to healthcare services between urban and rural settings, as well as highlight disparities in the needs and experiences of women with motor disabilities.

For Healthcare Professionals:

Profile: The strata will consist of healthcare professionals such as doctors, midwives, and nurses.

Place of Practice:Strata will be categorized into urban and rural practices.

Within each stratum, random sampling will be conducted by drawing lots from lists provided by organizations for persons with disabilities, the Ministry of Health and Social Protection's database of healthcare workers, rehabilitation center records, and other relevant sources.

## Sample size

**Quantitative phase.**   Based on the 2014 General Census of Population and Housing [17], the total population in the Rabat prefecture was 577,827, while Khémisset province had 542,221 people, of which 281,079 lived in urban areas and 261,142 in rural regions. According to the 2014 National Disability Survey (ENH2), 3.07% of the population had mobility disabilities, with 50.5% being women. From this data, the estimated population with mobility disabilities is calculated as follows:

Rabat: 8,885 individuals, Khémisset: 8,342 individuals, The sample size was determined using the Statcalc module from Epi-info software (v7.2.6.0), applying these parameters:

Estimated proportion (p): 0.5, to maximize sample size.

Confidence level (Z): 95%, corresponding to a Z-value of 1.96.

Margin of error (E): ±5% (0.05).

The calculated sample sizes are:Rabat: 384, rounded up to 400 women.Khémisset: 384, rounded up to 400 women, including 200 from rural areas to account for the factthat 48% of Khémisset's population is rural.Thus, the final sample sizes are:Rabat: 400 participants and Khémisset: 400 participants (200 urban, 200 rural).

For healthcare professionals, according to the health map, there are 541 primary care professionals in Rabat and Khémisset, with 133 in rural areas. Using similar sampling parameters, the sample size for healthcare professionals was set at 229, including 54 from rural areas, ensuring a balanced representation of both urban and rural providers involved in SRH services.

**Qualitative phase.**   For the qualitative phase, weaim to include a sample size ranging between 20 and 30 participants. However, recruitment will continue until data saturation is reached [18].

**Inclusion and exclusion criteria.**   Women with Physical Impairments:

The study will include women with diagnosed physical impairments affecting mobility or walking, as defined by the Centers for Disease Control and Prevention (CDC) [19]. The severity of the impairment—mild, moderate, or severe—will be assessed using a method that evaluates the extent to which a motor disability limits an individual's activities [20].Mild disability: The individual experiences some limitations but does not require assistance with Instrumental

**Table 1. Disabilitystaging system based on Activities of Daily Living (ADLs) and Instrumental Activities of Daily Living (IADLs).**

| Stage | ADL Domain | IADL Domain |
|---|---|---|
| Stage 0 : no disability | Can eat, toilet, dress, bath/shower, get in/out of bed or chairs, and walk without difficulty | Can use the telephone, manage money, prepare meals, do light house work, shop for personal items, and do heavy house work without difficulty |
| Stage 1 : mild disability | Eating, toileting, dressing, and bathing/ showring are no difficult ; may have difficulty getting in/out of bed or chairs and/ or walking | Using the telephone, managing money, preparing meals, and doing light house work are not difficult ;may have difficulty shopping for personal items and /or doing heavy housework |
| Stage 2 : moderate disability | Eating and toileting are nit difficult ; may have difficulty dressing, bathing/showering, getting in/out of bed or chairs,and /or walking | Using the telephone and managing money are not difficut ; may have difficulty preparing meals, doing light housework ; shopping for personal items and /or doing heavy house work |
| Stage 3 : severe disability | Difficulty with eating and/or toileting but not with all ADLs | Has difficulty using the telephone and/or managing money but not all IADLs are difficult |
| Stage 4 : complete disability | All ADLs are difficult | All IADLs are difficult |

Activities of Daily Living (IADLs) or Activities of Daily Living (ADLs), Moderate disability: The person requires assistance withIADLs but not with ADLs, Severe disability: The individual requires assistance with both IADLs and ADLs, Table 1 presents a disability staging system based on ADLs and IADLs.(**S1 Table**)

Additional criteria for inclusion include having a motor disability lasting for at least six months, residing in the prefecture of Rabat or the province of Khémisset, and being between the ages of 18 and 49.

Healthcare Professionals:

The study will also include healthcare professionals such as physicians, nurses, midwives, social workers, and others working in public health facilities (HPF) or regional centers for reproductive health (RCRH) in Rabat or Khémisset, providing sexual and reproductive health (SRH) services. To qualify, these professionals must have worked for more than one year in such facilities.

Decision-makers and Program Leaders;

Participants involved in decision-making or responsible for implementing strategies, policies, and programs related to SRH and disability will be included. This group consists of managers and leaders overseeing the operational execution of SRH and disability programs.

NGOs and Associations:

The study will include representatives from public-interest NGOs and associations active in the fields of disability or SRH, including those representing people with disabilities. Only organizations with a minimum of three years of experience in these areas will be considered.

**Exclusion criteria.** The study will exclude women with other types of disabilities or multiple disabilities,with cognitive impairments, outside the age range of 18–49,with temporary disabilities lasting less than six months,residing outside Rabat or Khémisset,unwilling to participate in the study.

## Data collection

**Qualitative data collection.** In-depth, semi-structured face-to-face interviews are the most suitable method for data collection in order to meet our research objectives. This

approach is ideal for capturing perceptions, attitudes, opinions, and beliefs, which are often better expressed through the reflective process facilitated by social interaction during an interview, as opposed to othermethods. The interview guides were developed based on a thorough review of the relevant literature [21–23], ensuring they elicit comprehensive qualitative data from participants.

The data collection tools will be carefully reviewed and validated by a team, including the Director of the Sexual and Reproductive Health Research Laboratory at Hassan University in Casablanca, the thesis advisor (agynecologist and Assistant Professor), a sociologist, and an Assistant Professor in Medical Informatics.

The principal investigator will conduct all interviews. For women with disabilities, interviews will be held in Moroccan dialectal Arabic, while other participants will have the option to choose between French or Arabic. The interviews are expected to last between 20 to 30 minutes, with flexibility to allow for breaks if needed.

For women with disabilities, interviews will be conducted in locations that ensure comfort and privacy. Healthcare professionals will be interviewed at their work places, maintaining confidentiality and a private setting. Other participants will be interviewed at their respective offices within their institutions or associations.

Before the main study, the interview guides will be pilot tested with five women with disabilities and five healthcare professionals in the Meknes province. This will help to ensure that the questions are clear and easy to understand.

All interviews willbe audio-recorded, translated, transcribed, and imported into NVivo software for analysis. Field notes will also be taken, focusing on the physical accessibility of Sexual and reproductive health services and reproductive health Services facilities, including the available infrastructure and equipment.

**Quantitative data collection.**   Quantitative data will be gathered through a structured survey, distributed to participants. The survey will encompass various areas, including demographic details, knowledge, attitudes, practices, and experiences related to sexual and reproductive health (SRH) services. The questionnaire will undergo pre-testing and refinement based on feedback received during this phase.

Two structured questionnaires have been developed, drawing on themes identified in the literaturereview [24, 25]. These focus on key indicators associated with SRH services, such as accessibility, service availability, satisfaction levels, encountered barriers, frequency of service use, and unmet needs.

The questionnaires will be administered in colloquial Arabicduring face-to-face interviews by the principal investigator (FZ), targeting participants whomeet the inclusion criteria. The translation into colloquial Arabic will be validated by the study supervisors to ensure accuracy.

For healthcare professionals, a self-administered version of the questionnaire will be available in both French and Arabic. It will be distributed in paperform, and also via Google Forms, particularly for healthcare professionals in rural areas where distributing and collecting paper questionnaires may be challenging.

The questionnaires will include several sections, covering sociodemographic information, personal attributes, knowledge, attitudes, practices, and perceptions related to the use of SRH services. Additionally, there will be a section for participants to provide recommendations and suggestions.

A pilot study will be conducted with 10 SRH service users and 5 healthcare professionals in the Meknes prefecture. This pilot will assess the clarity of the questionnaires and identify any necessary modifications. The anticipated completion time for each questionnaire is around 15 minutes.

## Data analysis

### Qualitative analysis

Data analysis will be carried out using NVivo software, following an inductive thematic analysis approach [26]. Audio recordings will be transcribed word-for-word, with the accuracy of the transcriptions carefully checked against the original interviews. These transcriptions will then be systematically coded, and the resulting codes will be organized into themes. To ensure consistency and relevance, the identified themes will be reviewed by both the thesis advisor and the professor specializing in medical informatics. Final themes, along with selected quotations, will be confirmed collaboratively with the supervisors. The themes will be presented along side the participants' socio-demographic data, such as gender, age, type of disability, occupation, and place of residence, using relevant textual quotes. The quality of the qualitative report will be evaluated using the Consolidated Criteria for Reporting Qualitative Research (COREQ) checklist [27].

### Quantitative data analysis

Quantitative data will undergo both descriptive and inferential statistical analysis. For the descriptive aspect, frequencies, means, and standard deviations will be computed. Aftercoding, the data will be entered, processed, and analyzed using R software. To examine the relation ships between access to and utilization of health services by women with motor disabilities and various independent variables, a bivariate analysis will be performed. Following this, a multivariate logistic regression will be applied to further explore and quantify the connections between the explanatory and dependent variables.

### Data protection

Confidential data will be securely stored on a restricted-access drive for five years. Prior to using the software analysis tool, all personally identifiable information will be removed from both transcriptions and audio recordings. Audio recordings will be destroyed immediately following transcription.

## Ethics and dissemination

### Ethical considerations

Ethical approval for this study was granted by the Ethics Committee for Biomedical Research (CERB) at Mohammed V University, Faculty of Medicine and Pharmacy in Rabat, Morocco (reference: ECBR 47–24, dated June 10, 2024), as well as by the Ethics Committee of Ibn Rochd University Hospital in Casablanca, Morocco (reference: 15/2024, dated July 22, 2024). Confidentiality and anonymity will be strictly upheld, with no personally identifiable information being collected. Informed consent will be obtained from all participants prior to conducting interviews or distributing questionnaires.

### Declaration of Helsinki

The study will prioritize maintaining the anonymity and confidentiality of all participants. Beforedistributing the studymaterials, participants willreceive a document outlining the study's objectives, the assurance of anonymity and confidentiality, and the potential benefits of participation. It will be emphasized that participation is entirely voluntary, and individuals may with draw from the study at any time without facing any consequences. Participants will be assured that their data will be used strictly for academic purposes, and their identities will

remain confidential. Verbal informed consent will be obtained from all participants. Furthermore, the study will comply with the ethical standards set forth in the Declaration of Helsinki for research involving humansubjects [28], along side obtaining necessary ethical approval.

## Limitations

This study's strength lies in its use of a mixed-methods approach, facilitating the triangulation of quantitative and qualitative research findings. This methodology provides a comprehensive understanding of the accessibility and utilization of health services by persons with disabilities (PWDs), while also offering insights intotheir perceptions and the functionality of these services. The inclusion of respondents from both rural and urban areas further enhances the study's ability to generalize its findings across different contexts. However, the research will be limited to a single region and will focus exclusively on individuals with physical disabilities. Additionally, the reliance on self-reported data may introduce social desirability and recall biases. Nonetheless, the triangulation of results will help mitigate these potential limitations.

## Expected outcomes

This study seeks to accomplish several outcomes by improving understanding of the challenges and enablers related to accessing sexual and reproductive health (SRH) services for women with physical disabilitie, formulating targeted recommendations to enhance the inclusiveness and accessibility of SRH services, raising awareness among healthcare providers and policymakers about the specific needs of women with motor disabilities and contributing valuable evidence to inform the development of inclusive healthpolicies and programs.

## Discussion

This study seeks to address a critical gap in research concerning the accessibility and use of sexual and reproductive health (SRH) services by women with motor disabilities. Understanding the distinct barriers and facilitators faced by this group is essential for developing effective interventions. The recommendations resulting from this research will have practical implications for improving the inclusiveness and accessibility of SRH services. By increasing awareness among healthcare professionals and policymakers about the specific needs of women with motor disabilities, the study aims to contribute to the development of more inclusive health programs and policies.

By employing a mixed-methods approach, the study will combine quantitative and qualitative data, allowing for a comprehensive understanding of the challenges and opportunities in this area. This method strengthens the validity of the findings while offering deeper insights that might be missed using a single approach. The researchis expected to uncover significant barriers to SRH services for women with motor disabilities, such as physical inaccessibility, lack of tailored services, and cultural stigma. These findings will be contextualized within both global research on disability and healthcare access, and the Moroccan context, including infrastructure and specific socio-cultural determinants.

The study seek also to propose several practical recommendations, emphasizing the need for policy reforms to ensure that SRH services are both inclusive and accessible.

## Conclusion

This study aims to provide insights into the barriers and facilitators affecting access to sexual and reproductive health services for women with motor disabilities in Rabat and Khemisset. By understanding the perspectives of both women with disabilities and healthcare

professionals, the study will contribute to developing recommendations for inclusive and equitable SRH services, ultimately improving health outcomes and promoting the rights of women with disabilities.

## Supporting information

**S1 Table. Disability staging system.** ADL and IADL Domains.
(DOCX)

## Author Contributions

**Conceptualization:** Firdaous Zekaoui.

**Methodology:** Firdaous Zekaoui.

**Project administration:** Houssine Boufettal.

**Resources:** Firdaous Zekaoui.

**Supervision:** Rachid Aboutaieb.

**Validation:** Houssine Boufettal, Rachid Aboutaieb.

**Writing – original draft:** Firdaous Zekaoui.

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
