## [Decision Letter · Decision Letter 0]

26 Aug 2024

PONE-D-24-06353Protocol for a qualiquantitative study of accessibility of sexual and reproductive health services among women with motor  disabilities in Morocco.PLOS ONE

Dear Dr. zekaoui,

Thank you for submitting your manuscript to PLOS ONE. After careful consideration, we feel that it has merit but does not fully meet PLOS ONE’s publication criteria as it currently stands. Therefore, we invite you to submit a revised version of the manuscript that addresses the points raised during the review process.

We look forward to receiving your revised manuscript.

Kind regards,

Mohamed Amine Baba

Academic Editor

PLOS ONE

Journal Requirements:

4. Please include a copy of Table 1 which you refer to in your text on page 14.

Reviewers' comments:

Reviewer's Responses to Questions

**Comments to the Author**

1. Does the manuscript provide a valid rationale for the proposed study, with clearly identified and justified research questions?

Reviewer #1: Partly

Reviewer #2: Yes

2. Is the protocol technically sound and planned in a manner that will lead to a meaningful outcome and allow testing the stated hypotheses?

Reviewer #1: Partly

Reviewer #2: Yes

3. Is the methodology feasible and described in sufficient detail to allow the work to be replicable?

Reviewer #1: Yes

Reviewer #2: Yes

4. Have the authors described where all data underlying the findings will be made available when the study is complete?

Reviewer #1: Yes

Reviewer #2: Yes

5. Is the manuscript presented in an intelligible fashion and written in standard English?

Reviewer #1: Yes

Reviewer #2: Yes

6. Review Comments to the Author

You may also provide optional suggestions and comments to authors that they might find helpful in planning their study.

Reviewer #1: REVISIONS TO BE MADE:

The subject matter is relevant and interesting, and requires thorough research. As we all know, the abstract is a key element for readers, search engines and databases. 1. So you'll need to reduce it by at least 100 to 250 words 2. Clarify your research question.

In the materials and methods section, it can be written in several stages:

- as soon as the research is conceived and the tests set up - General characteristics of the study, variables measured, methods, techniques, measuring and measurement and observation equipment

- during research - Protocol adaptation, unforeseen events

- to data analysis - Statistical processing.

Avoid :

- elements not directly related to the study objective

- results, except in the case of method validation results on a prior sample

- sample

- the omission of any element enabling the experiment to be reproduced.

In the discussion, you explain your results, i.e. you give their scientific meaning.

You need to expand the Discussion section.

Reviewer #2: Here are some comments about the study protocol:

Relevance and Importance:

The study addresses a highly significant and often overlooked issue-access to sexual and reproductive health (SRH) services for women with physical disabilities. By focusing on Morocco, the research fills a gap in the literature regarding the specific barriers and facilitators in a non-Western context, making it highly relevant for both local and global health agendas.

Clear Objectives:

The abstract clearly outlines the study's objectives, particularly its aim to identify the barriers and facilitators to SRH services for women with disabilities. This clarity is essential for ensuring that the study's findings can be effectively translated into actionable recommendations.

Methodological Rigor:

The use of a mixed-methods approach strengthens the study by allowing for both quantitative and qualitative data collection. This dual approach ensures a comprehensive understanding of the issue, combining statistical analysis with in-depth personal insights from the affected population and healthcare professionals.

Innovative Aspects:

The recruitment strategy, which involves collaboration with NGOs, local organizations, and social media campaigns, is innovative and practical. It reflects a commitment to inclusivity and ensures a diverse sample that represents both urban and rural populations.

Potential Impact:

The study's potential to influence policy and practice is significant. By proposing recommendations based on empirical data, the research could lead to more inclusive and equitable SRH services in Morocco, directly impacting the health and rights of women with disabilities.

Challenges and Limitations:

While the study is well-conceived, there may be challenges in ensuring the accurate representation of the population, particularly in rural areas where access to participants might be limited. Additionally, the reliance on self-identification for inclusion criteria could introduce bias, as individuals who do not perceive themselves as disabled may be excluded.

Ethical Considerations:

The abstract mentions strict confidentiality measures, which is critical given the sensitive nature of SRH services. However, it would be beneficial to see more detail on how informed consent will be obtained, particularly for participants who may have communication or cognitive difficulties.

Generalizability:

While the study is focused on specific regions in Morocco, its findings could have broader implications for other regions and countries with similar cultural and healthcare contexts. However, the unique sociocultural factors in Morocco should be carefully considered when generalizing the results to other settings.

Future Directions:

The study sets a strong foundation for future research, particularly in exploring how the proposed recommendations are implemented and their impact on SRH services over time. Longitudinal studies could be valuable in assessing the long-term outcomes of the interventions proposed.

Contribution to Global Health:

The research aligns well with global health priorities, including the Sustainable Development Goals (SDGs). By focusing on the intersection of disability and SRH, the study contributes to the broader discourse on health equity, making it a valuable addition to global health literature.

7. PLOS authors have the option to publish the peer review history of their article (what does this mean?). If published, this will include your full peer review and any attached files.

Reviewer #1: No

Reviewer #2: **Yes: **Dr Soufiane Bigi

---

## [Author Response · Author response to Decision Letter 0]

17 Sep 2024

Response to the editor

Dear Dr. Baba,

Thank you for taking the time to review my manuscript and providing thoughtful feedback. I appreciate the opportunity to revise and strengthen the paper in line with the journal's standards. I have carefully reviewed your comments and the reviewers’ suggestions. Likewise, I have made the necessary revisions to address each point raised and ensure the manuscript aligns with the journal’s guidelines :

I have considered your comments and revised the manuscript to align with PLOS ONE’s style guidelines. This includes reformatting the document according to the provided templates and making necessary adjustments, such as correcting file names.

We would like to clarify that there are no specific ethical or legal restrictions on sharing our dataset. The data will be anonymized to protect the confidentiality of the participants, and no personal identifying information will be shared. There have been no additional restrictions from any ethics committee or institutional body. For data requests or further inquiries, we are available to provide any necessary information while adhering to confidentiality and ethical guidelines.

We have included a full ethics statement in the Method section, as per your suggestion.

We have now included Table 1, which is referenced on page 14.

I have carefully reviewed and corrected the reference list to ensure that it is both complete and accurate.

 Thank you for the detailed and constructive feedback provided for my manuscript. I truly appreciate the time and effort you and the reviewers have dedicated to improving the quality of the paper.

Attached my response to the comments of reviewers, please let me know if any further revisions are required. I appreciate your continued guidance and look forward to your feedback.

Sincerely,

Response to the reviewers 

Reviewer #1

Thank you for your insightful feedback on my manuscript. I appreciate the time and effort you put into reviewing it and providing constructive suggestions. Below, I address your comments point by point:

Comment : Abstract length

Response : I have reduced the abstract by approximately 150 words, ensuring it remains concise while retaining key information for both readers and search engines.

Comment : Research question

Response : I have refined the research question to make it clearer and easier to identify, providing a precise summary of the research objective at the beginning of the abstract. The primary research question is now explicitly stated, followed by the secondary questions.

Comment :Materials and Methods section

Response : I have restructured this section based on your suggestions, detailing the process in several stages. This includes:

o Study characteristics: A comprehensive overview of the study design, including variables measured, methodologies, and the equipment used.

o Research adaptations: I have added details on any adjustments made to the research protocol and highlighted any unforeseen events that occurred. These revisions ensure the study is reproducible by other researchers.

o 

Comment : Discussion section expansion

Response : Following your recommendation, I have expanded the discussion to elaborate on the expected outcomes, potential limitations, and future research directions.

Once again, thank you for your valuable feedback. I believe these revisions have strengthened the manuscript and look forward to your further comments.

Reviewer #2

Thank you for your insightful feedback on my manuscript. I appreciate the time and effort you put into reviewing it and providing constructive suggestions and sincerely appreciate your thoughtful review. Below, I address each of the points you raised:

Comment : Relevance and Importance

Response : Thank you for acknowledging the importance of the study on access to sexual and reproductive health (SRH) services for women with physical disabilities in Morocco. Given the lack of research in non-Western contexts, we aim to fill this gap and contribute to global efforts in ensuring equitable access to SRH services, particularly for marginalized populations like women with disabilities.

Comment : Clear Objectives

Response : We structured the abstract to clearly outline the study's objective, which is to identify the barriers and facilitators women with disabilities encounter when seeking SRH services. Our goal is to produce findings that are both informative and actionable, with the potential to influence policy and practice.

Comment : Methodological Rigor

Response : We appreciate your recognition of the strength in our mixed-methods approach. By integrating quantitative and qualitative data, we aim to provide a comprehensive understanding of the issue. The combination of statistical data and personal narratives offers a deep exploration of the challenges faced by these women, along with healthcare professionals' perspectives on how these can be addressed.

Comment : Innovative Aspects

Response : We are encouraged by your acknowledgment of our recruitment strategy, which includes collaboration with NGOs, local organizations, and the use of social media to reach a diverse and inclusive participant pool. This approach will enable us to capture experiences from both urban and rural settings.

Comment : Potential Impact

Response : We are grateful for your recognition of the study's potential to influence policy both locally and globally. Our goal is to use empirical data to advocate for more inclusive SRH services and to provide recommendations that reflect the lived experiences of women with disabilities.

Comment : Challenges and Limitations

Response : The reviewer’s observation regarding potential challenges in reaching rural populations and the reliance on self-identification for inclusion criteria is well-founded. We acknowledge that these are limitations we have considered. In response, we will made efforts to ensure robust recruitment strategies in rural areas, working closely with local organizations familiar with these communities. Additionally, we aim to mitigate bias by employing flexible inclusion criteria to include with diagnosed physical impairments affecting mobility or walking, as defined by the Centers for Disease Control and Prevention and ensuring that our sampling captures a wide range of experiences

Comment 7 :Ethical Considerations

Response :We have carefully considered your point about ethical concerns. A verbal informed consent process will be implemented to accommodate participants with varying literacy levels, ensuring their full understanding of the study. Participants with significant cognitive impairments will be excluded to maintain the ethical integrity of the study, though we recognize the need for future research that includes such individuals in an ethical and inclusive manner.

Comment : Generalizability

Response : While our findings are specific to Morocco, we agree that themes of accessibility and inclusion are broadly applicable across various cultural contexts. We hope that our research will inspire further studies in other low- and middle-income countries.

Comment : Future Directions

Response :We appreciate your acknowledgment of the study's potential for future research. We envision this work as a foundation for longitudinal studies to assess the impact of our recommendations over time, focusing on the evolution of SRH services in response to our findings.

Comment : Contribution to Global Health

Response : It is gratifying to see that our study aligns with global health priorities, particularly the Sustainable Development Goals (SDGs). By addressing the intersection of disability and SRH, we hope to contribute to the ongoing discourse on health equity and inclusive health systems.

Once again, thank you for your valuable feedback. I believe these revisions have strengthened the manuscript and look forward to your further comments.

---

## [Editor Report · Decision Letter 1]

23 Sep 2024

Protocol for a qualiquantitative study of accessibility of sexual and reproductive health

services among women with motor disabilities in Morocco.

PONE-D-24-06353R1

Dear author ,

We’re pleased to inform you that your manuscript has been judged scientifically suitable for publication and will be formally accepted for publication once it meets all outstanding technical requirements.

Kind regards,

Mohamed Amine Baba

Academic Editor

PLOS ONE
---

## [Editor Report · Acceptance letter]

17 Oct 2024

PONE-D-24-06353R1 

PLOS ONE

Dear Dr. zekaoui, 

I'm pleased to inform you that your manuscript has been deemed suitable for publication in PLOS ONE. Congratulations! Your manuscript is now being handed over to our production team.

Kind regards, 

on behalf of

Pr Mohamed Amine Baba 

Academic Editor

PLOS ONE